# Probing the ionotropic activity of glutamate GluD2 receptor in HEK cells with genetically-engineered photopharmacology

Damien Lemoine[1], Sarah Mondoloni[1], Jérome Tange[1], Bertrand Lambolez[1], Philippe Faure[1], Antoine Taly[2,3†], Ludovic Tricoire[1†*], Alexandre Mourot[1†*]

[1]Neuroscience Paris Seine - Institut de Biologie Paris Seine (NPS - IBPS), CNRS, INSERM, Sorbonne Université, Paris, France; [2]CNRS, Université de Paris, UPR 9080, Laboratoire de Biochimie Théorique, Paris, France; [3]Institut de Biologie Physico-Chimique-Fondation Edmond de Rothschild, PSL Research University, Paris, France

*For correspondence:
ludovic.tricoire@upmc.fr (LT);
alexandre.mourot@inserm.fr (AM)

[†]These authors contributed equally to this work

Competing interests: The authors declare that no competing interests exist.

**Abstract** Glutamate delta (GluD) receptors belong to the ionotropic glutamate receptor family, yet they don't bind glutamate and are considered orphan. Progress in defining the ion channel function of GluDs in neurons has been hindered by a lack of pharmacological tools. Here, we used a chemo-genetic approach to engineer specific and photo-reversible pharmacology in GluD2 receptor. We incorporated a cysteine mutation in the cavity located above the putative ion channel pore, for site-specific conjugation with a photoswitchable pore blocker. In the constitutively open GluD2 Lurcher mutant, current could be rapidly and reversibly decreased with light. We then transposed the cysteine mutation to the native receptor, to demonstrate with high pharmacological specificity that metabotropic glutamate receptor signaling triggers opening of GluD2. Our results assess the functional relevance of GluD2 ion channel and introduce an optogenetic tool that will provide a novel and powerful means for probing GluD2 ionotropic contribution to neuronal physiology.

## Introduction

The delta glutamate receptors, GluD1 and GluD2, belong to the ionotropic glutamate receptor (iGluR) family, yet they don't bind glutamate (*Yuzaki and Aricescu, 2017*). They are considered as glutamate receptors solely based on their strong sequence and structure homology with AMPA, NMDA and kainate receptors (*Lomeli et al., 1993*; *Araki et al., 1993*; *Schmid and Hollmann, 2008*; *Elegheert et al., 2016*; *Burada et al., 2020a*; *Burada et al., 2020b*). GluD receptors are widely expressed throughout the brain, GluD1 predominantly in the forebrain, while GluD2 is highly enriched in cerebellar Purkinje cells (PCs) (*Konno et al., 2014*; *Hepp et al., 2015*; *Nakamoto et al., 2020b*). Both GluD1 and GluD2 play a role in the formation, stabilization, function and plasticity of synapses through their interaction with members of the cerebellin (Cbln) family (*Fossati et al., 2019*; *Tao et al., 2018*; *Matsuda et al., 2010*; *Kakegawa et al., 2008*). Cbln1 notably binds both the N-terminal domain of postsynaptic GluD2 and presynaptic neurexins, leading to a trans-synaptic bridge that promotes synaptogenesis and is essential for GluD2 signaling in vivo (*Elegheert et al., 2016*; *Suzuki et al., 2020*). Deletion of genes coding for GluD1 or GluD2 in mouse results in marked behavioral alterations (*Yadav et al., 2012*; *Lalouette et al., 2001*; *Yadav et al., 2013*; *Nakamoto et al., 2020a*), and mutations in these genes in humans have been associated with neuro-developmental and psychiatric diseases (*Griswold et al., 2012*; *Treutlein et al., 2009*; *Greenwood et al., 2011*; *Cristino, 2019*), attesting to their functional importance in brain circuits.

**eLife digest** Neurotransmitters are chemicals released by the body that trigger activity in neurons. Receptors on the surface of neurons detect these neurotransmitters, providing a link between the inside and the outside of the cell. Glutamate is one of the major neurotransmitters and is involved in virtually all brain functions. Glutamate binds to two different types of receptors in neurons. Ionotropic receptors have pores known as ion channels, which open when glutamate binds. This is a fast-acting response that allows sodium ions to flow into the neuron, triggering an electrical signal. Metabotropic receptors, on the other hand, trigger a series of events inside the cell that lead to a response. Metabotropic receptors take more time than ionotropic receptors to elicit a response in the cell, but their effects last much longer.

One type of receptor, known as the GluD family, is very similar to ionotropic glutamate receptors but does not directly respond to glutamate. Instead, the ion channel of GluD receptors opens after being activated by glutamate metabotropic receptors. GluD receptors are produced throughout the brain and play roles in synapse formation and activity, but the way they work remains unclear. An obstacle to understanding how GluD receptors work is the lack of molecules that can specifically block these receptors' ion channel activity.

Lemoine et al. have developed a tool that enables control of the ion channel in GluD receptors using light. Human cells grown in the lab were genetically modified to produce a version of GluD2 (a member of the GluD family) with a light-sensitive molecule attached. In darkness or under green light, the light-sensitive molecule blocks the channel and prevents ions from passing through. Under violet light, the molecule twists, and ions can flow through the channel.

With this control over the GluD2 ion channel activity, Lemoine et al. were able to validate previous research showing that the activation of metabotropic glutamate receptors can trigger GluD2 to open. The next step will be to test this approach in neurons. This will help researchers to understand what role GluD ion channels play in neuron to neuron communication.

Despite their structural similarity with other iGluRs, and notably the presence of a ligand-binding domain (LBD), GluDs stand out because they are not activated by glutamate (*Araki et al., 1993*; *Lomeli et al., 1993*). Nonetheless, recent studies revealed that GluD pore opening could be triggered indirectly, through the activation of Gq-coupled metabotropic receptors, and contributes to neurotransmission and neuronal excitability (*Ady et al., 2014*; *Dadak et al., 2017*; *Benamer et al., 2018*; *Gantz et al., 2020*). Indeed, the activation of GluD channels triggered by metabotropic glutamate receptors (mGlu1/5) underlies the slow excitatory postsynaptic current in cerebellar PCs (GluD2, *Ady et al., 2014*) and in midbrain dopaminergic neurons (GluD1, *Benamer et al., 2018*). Moreover, the demonstration that GluD1 channels carry the noradrenergic slow excitatory current in dorsal raphe neurons (*Gantz et al., 2020*), suggests that the contribution of GluD channels to neuronal excitability and synaptic physiology may be widespread. The above studies relied largely on genetic tools, such as dead-pore mutants or targeted deletions, to assess the ion channel function of GluDs. Yet, due to the absence of specific pharmacological tools to block their ion permeation, the role of GluD1/2 channels in the regulation of neural activity remains largely elusive.

Pore blockers for GluDs, such as pentamidine and 1-Naphthyl acetyl spermine (NASPM), were previously identified using a point mutation (A654T) in GluD2 that confers constitutive ion flow and causes the degeneration of cerebellar PCs in *Lurcher* (GluD2$^{Lc}$) mice (*Wollmuth et al., 2000*; *Zuo et al., 1997*). These molecules are however also pore blockers of NMDA and AMPA receptors, respectively. Other ligands such as D-serine and glycine bind to the LBD and reduce spontaneous currents in GluD2$^{Lc}$, which suggests a coupling between the LBD and the channel (*Naur et al., 2007*; *Hansen et al., 2009*), but these molecules have broad spectrum activity. Finally, 7-chloroky-nurenic acid has been identified to modulate GluD2$^{Lc}$ current by binding to the D-serine site but it is also a GluN1 competitive antagonist (*Kristensen et al., 2016*).

To fill this gap, we bestowed light-sensitivity to the GluD ion channel pore using a photoswitchable tethered ligand (PTL) approach (*Paoletti et al., 2019*; *Mondoloni et al., 2019*). Using structure-based design, we incorporated a cysteine point mutation at the surface of GluD2, right above the hypothetical channel lumen, onto which can be anchored a photoswitchable pore blocker.

Different wavelengths of light are then used to modify the geometry of the PTL, thereby presenting/ removing the blocker to/from the channel, resulting in optical control of ionotropic activity. Here we demonstrate rapid and reversible, optical control of ion current through a cysteine-substituted GluD2 receptor. This novel tool, called light-controllable GluD2 (LiGluD2), allows rapid, reversible and pharmacologically-specific control of ionic current through GluD2, and may help provide a mechanistic understanding of how this receptor contributes to brain circuit function and behaviors.

## Results

### Designing a light-controllable GluD receptor

Our approach to probing the functionality of the ion channel in GluD is to install a photo-isomerizable pore blocker at the extracellular entrance to the channel lumen (*Figure 1A*). The tethered ligand is site-specifically attached to a cysteine-substituted residue. In darkness or under green light (500–535 nm), the PTL adopts an elongated shape and reaches the lumen, resulting in ion channel blockade, while under violet light (380–390 nm), it switches to a twisted, shorter configuration, relieving blockade. Our design of the PTL was based on the chemical structure of pentamidine (*Figure 1B*), a pore blocker that efficiently blocks current through GluD2^Lc receptors (*Williams et al., 2003*). The PTL, called MAGu, contains a thiol-reactive maleimide (M) moiety, a central photo-isomerizable azobenzene (A) chromophore, and a guanidinium (Gu) head group that resembles the amidinium groups of pentamidine (*Figure 1C*). MAGu was selected notably because its synthesis route has been described (referred to as PAG1c in the original article) and because it was shown to have no adverse effect on native brain tissue (*Lin et al., 2015*). In aqueous solution, MAGu could be

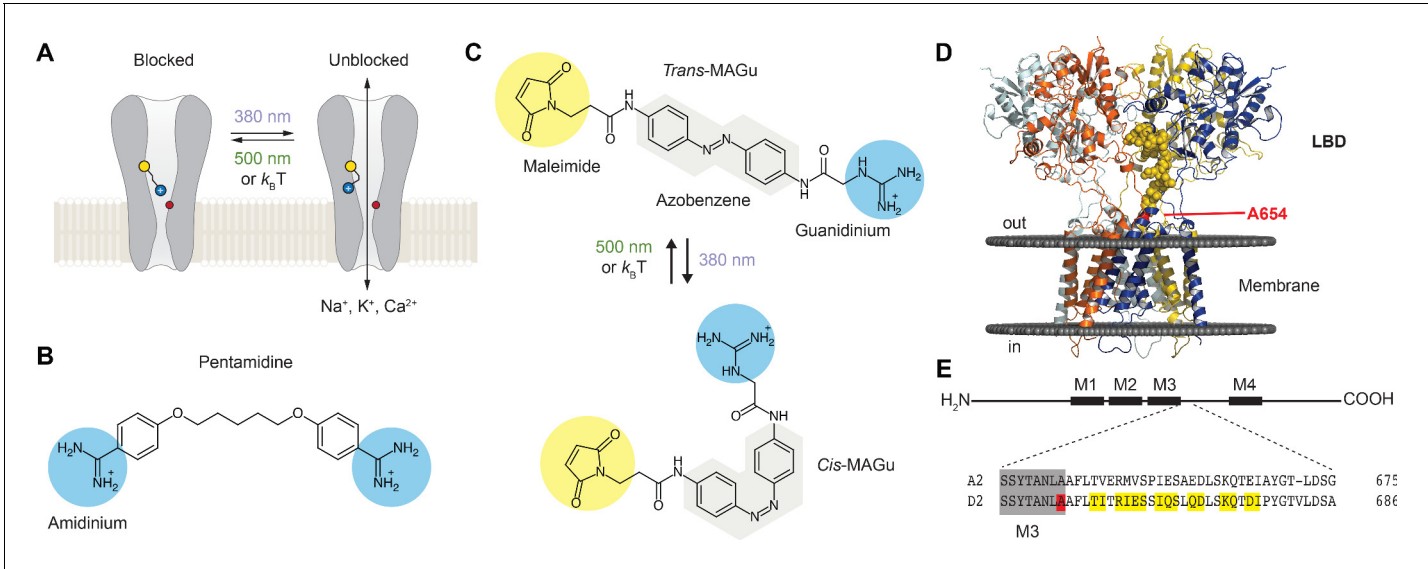

**Figure 1.** Optogenetic pharmacology strategy to probe the ionotropic activity of GluD receptors. (A) GluD2 is genetically-modified to incorporate a cysteine residue (yellow) at the entrance to the pore, which serves as a handle for the covalent attachment of a synthetic, photoswitchable tethered ligand (PTL). Under green light (500 nm), the PTL adopts an elongated state and places its cationic head group in the lumen, resulting in ion channel blockade. Under violet light (380 nm), the PTL switches to a twisted, shorter form and unblocks the channel. The position of the Lurcher (Lc) mutation, which produces a permanently open channel, is depicted in red. (B) Chemical structure of pentamidine, a non-selective iGluR blocker with two amidinium head groups. (C) Chemical structures of the PTL MAGu in its *trans* (top) and *cis* (bottom) configurations. MAGu is composed of a cysteine-reactive maleimide group, a central azobenzene chromophore, and a guanidinium cationic head group. (D) Molecular model of GluD2, based on the structure of activated GluA2 (5weo). Residues mutated to cysteine are depicted in yellow, while the Lc mutant is shown in red. (E) Top, schematic representation of one GluD subunit, with its ligand-binding domain (LDB) and its four membrane segments (M1-4, M2 being a non-membrane spanning pore loop). Bottom, sequence alignment between the mouse GluA2 and GluD2 receptors around the engineered mutations. M3 is shown in gray, the 15 residues mutated to cysteine in yellow, and the position of the A654T Lc mutation in red.

The online version of this article includes the following figure supplement(s) for figure 1:

**Figure supplement 1.** Photochemical properties of MAGu.

converted to its *cis* form using 380 nm light, and converted back to *trans* either slowly in darkness ($t_{1/2}$ ~ 20 min) or rapidly upon illumination with 525 nm light (*Figure 1—figure supplement 1A–B*), in agreement with previous reports (*Lin et al., 2018*). To find the best attachment site for MAGu on GluD, we developed a homology model of the GluD2 receptor, based on the structure of the recently crystallized GluA2 receptor (*Twomey et al., 2017*) (see methods). Using this model, we selected a series of 15 residues, located on the peptide that links the LBD to the third transmembrane domain (M3) that lines the channel lumen, for mutation to cysteine (*Figure 1D–E*).

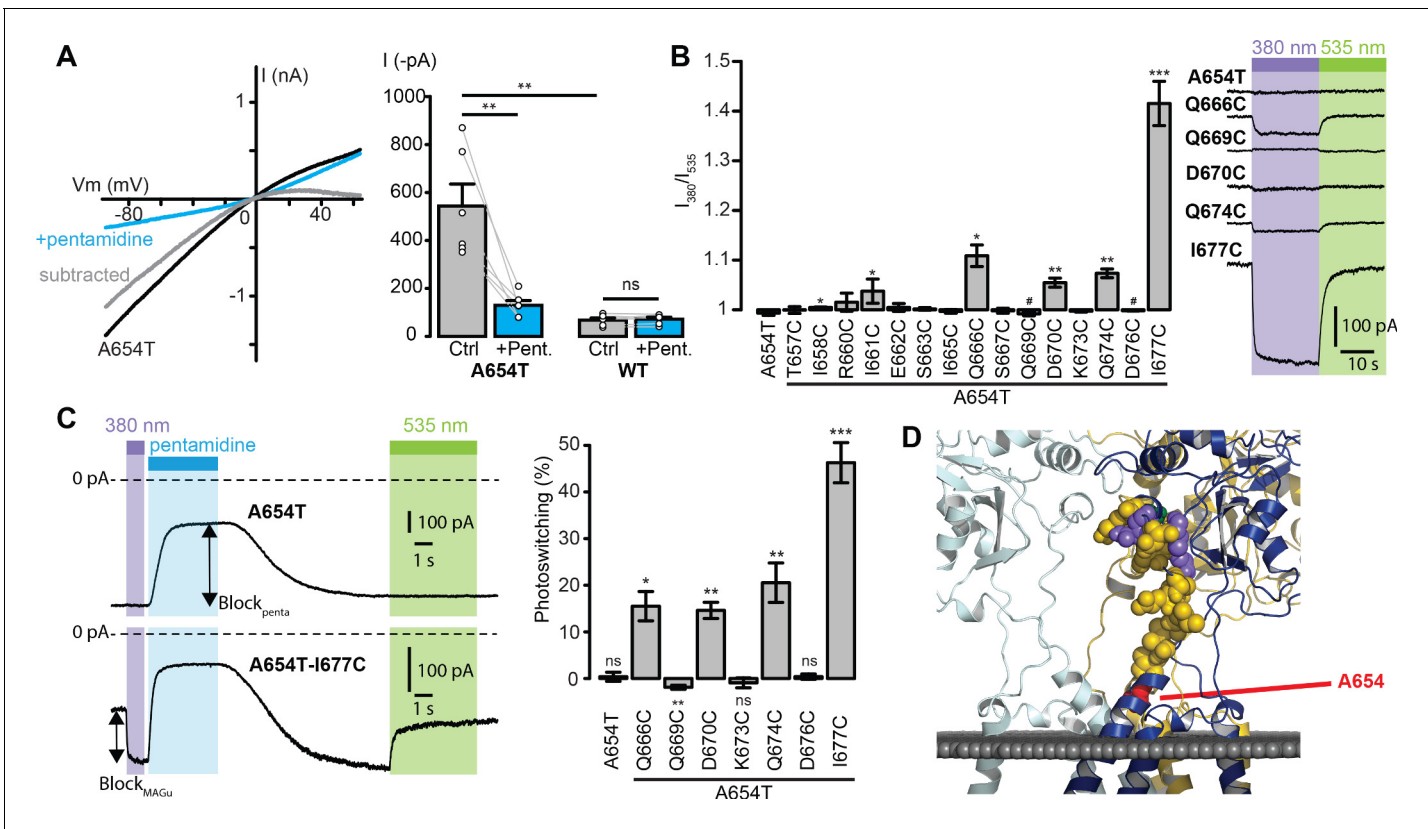

**Figure 2.** Screening of the fifteen single-cysteine mutants engineered on the Lc background. (**A**) Left, representative current-voltage relationship for GluD2-A654T Lc, with (blue) and without (black) pentamidine (100 µM). The subtracted current (gray) shows clear inward rectification and block at positive voltages. Right, currents recorded at −60 mV were larger for GluD2-A654T than for WT (n = 6 cells, p=0.0033, two-sample t-test), and were strongly reduced with 100 µM pentamidine for A654T (n = 6 cells, p=0.0045, paired t-test) but not for the WT GluD2 (n = 6 cells, p=0.34, paired t-test). (**B**) Left, ratio of the currents recorded at −60 mV under 380 and 535 nm light, for A654T and the fifteen cysteine mutants engineered on the A654T background (n = 3–8 cells, one-sample t-test, or Wilcoxon when normality is not verified, compared with a theoretical mean value of 1). Right, representative change in holding current when switching between dark, 380 and 535 nm light, for A654T and for five mutants that show modulation of the holding current when switching between 380 and 535 nm light (p<0.05 except for Q669C where p=0.09). (**C**) Left, representative current traces (Vm = −60 mV) recorded for A654T and A654T-I677C in darkness, then under 380 nm light (violet), upon application of pentamidine (blue, 100 µM), and then under 535 nm after pentamidine washout (green). Right, percent photoswitching (PS = Block_photo/Block_penta) for A654T and the cysteine mutants that show both clear leak current and pentamidine block. Photoswitching reached 15.5 ± 3.1% for Q666C (p=0.016, n = 4),–1.9 ± 0.4% for Q669C (p=0.008, n = 6), 14.6 ± 1.7% for D670C (p=0.003, n = 4), 20.5 ± 4.2% for Q674C (p=0.008, n = 5) and 46.3 ± 4.3% for I677C (p=1.34E-05, n = 8, one-sample t-test, or Wilcoxon when normality is not verified, compared with a theoretical mean value of 0). Photoswitching was absent in the control A654T mutant (0.37 ± 0.9%, p=0.72, n = 5) and in the other two cysteine mutants K673C (−0.9 ± 1.06%, p=0.43) and D676C (0.39 ± 0.57%, p=0.53). (**D**) Model of GluD2 showing the location of A654 (red), Q666, D670, Q674 and I677C (violet), Q669 (green), and all the non-photocontrollable cysteine mutants (yellow). Data are presented as mean value ± sem. Source files of individual data points used for the quantitative analysis are available in the *Figure 2—source data 1*.

The online version of this article includes the following source data and figure supplement(s) for figure 2:

**Source data 1.** Related to *Figure 2A, B and C*.

**Figure supplement 1.** Functional characterization of the cysteine mutants.

### Cysteine screening

Since no known ligand directly gates the ion channel of GluD2, we used a Lc mutant, A654T, which displays a constitutively open channel (*Wollmuth et al., 2000*; *Zuo et al., 1997*), for screening the 15 single-cysteine mutations. Accordingly, we found that heterologous expression of GluD2-A654T, but not of the wild-type (WT) protein, in HEK cells produces large currents that reverse at membrane potential close to 0 mV and are reduced by externally-applied pentamidine (*Figure 2A*). Subtracted Lc current showed clear rectification at positive potentials, as reported with the blockade by NASP, another GluD blocker (*Kohda et al., 2000*). Therefore, the A654T Lc mutant was subsequently used as a screening platform to find the best attachment site for MAGu on GluD2. Each of the 15 residues identified in *Figure 1D* were mutated individually to cysteine on the A654T background, and tested using patch-clamp electrophysiology. Cells were treated with MAGu (20 µM, 20 min) and Lc currents were measured in voltage-clamp mode (−60 mV) under different illumination conditions to toggle MAGu between its *cis* and *trans* states. As expected, current through A654T was not affected by light, indicating that in the absence of a properly-positioned cysteine, MAGu has no effect on this Lc channel. In contrast, we found several cysteine mutants for which current was significantly larger under 380 than under 535 nm light, and one mutant (Q669C) for which there was a tendency for 'reverse photoswitching', that is larger currents under 535 than under 380 nm light (*Figure 2B*). We

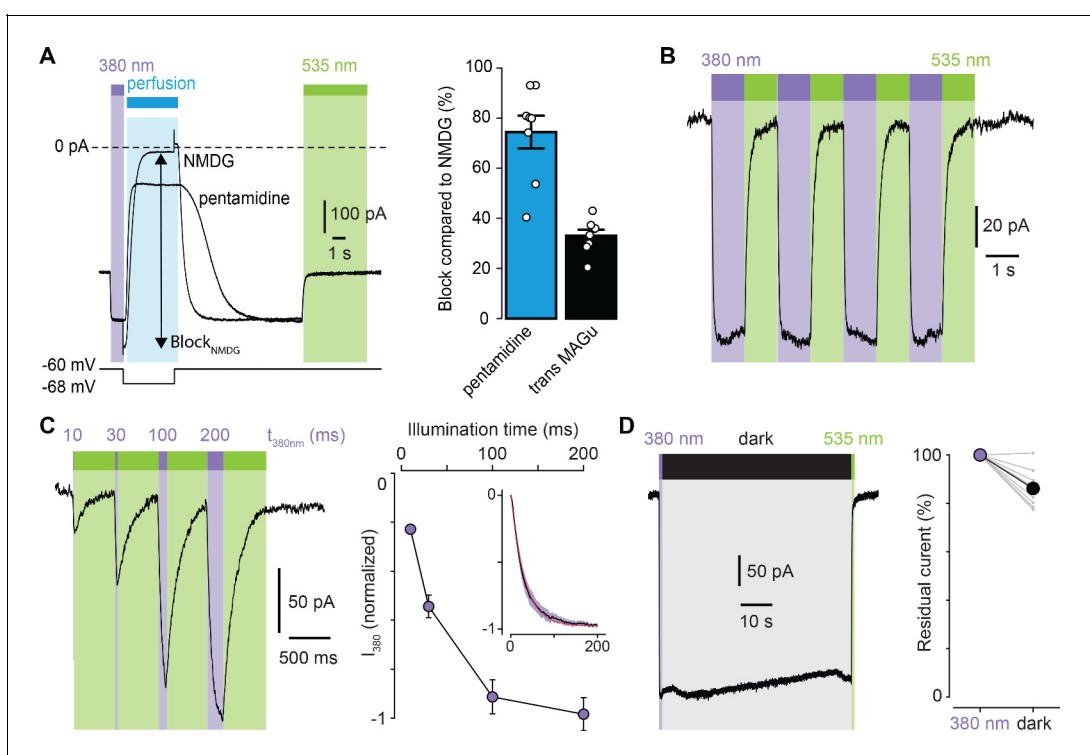

**Figure 3.** Photoregulation of GluD2-A654T-I677C labeled with MAGu. (**A**) Representative current traces (Vm = −60 mV) recorded for A654T-I677C in darkness, then under 380 light (violet), upon application of pentamidine or NMDG (blue, 100 µM), and then under 535 nm (green). Membrane potential is switched from −60 to −68 mV during NMDG application to correct for the change in junction potential, and percent Block$_{NMDG}$ is calculated for Vm = −68 mV. Right, percent block for pentamidine (blue, 74.4 ± 6.5%) and for transMAGu (green, 535 nm, 33.0 ± 2.4%) compared to NMDG block (n = 8 cells). (**B**) Representative recording showing the reversibility of block/unblock over multiple cycles of 380/535 nm light. (**C**) Left, representative recording showing the extent of current unblock when varying the illumination time under violet light. Right, quantification of current unblock as a function of illumination time (n = 8 cells). Inset, averaged time-course of current unblock when switching from dark to 380 nm light (mean value in black, ± SEM in purple, n = 7 cells) and corresponding mono-exponential fit (red, k = 0.0296 ± 0.0002 ms$^{-1}$; t$_{1/2}$ = 23.4 ms). (**D**) Left, representative current trace showing the thermal stability of *cis* MAGu in darkness after a brief flash of 380 nm light. Right, 86.2 ± 2.5% of the residual current remains after 1 min in darkness (n = 9 cells). Data are presented as mean value ± sem. Source files of individual data points used for the quantitative analysis are available in the *Figure 3—source data 1*.

The online version of this article includes the following source data for figure 3:

**Source data 1.** Related to *Figure 3A, C and D*.

then quantified the degree of photoswitching by comparing the block in darkness (*trans* state) to the block evoked by a saturating concentration of pentamidine (100 µM). We excluded from the analysis mutants that displayed no pentamidine-decreased leak current (i.e. mutants for which pentamidine block was significantly smaller than that observed on A654T, *Figure 2—figure supplement 1A–B*), because they were likely either not expressed or not functional. Photoswitching was significant for Q666C, Q669C, D670C, Q674C, and I677C, suggesting that MAGu covalently reacted with these cysteine mutants and that, once tethered, it could modulate current in one of its conformer (*Figure 2C*). Importantly, photomodulation was absent in the control A654T and the other cysteine mutants, indicating that the effect of light is specific to the attachment of MAGu to the above-mentioned cysteine mutants. From a structural point of view, the photocontrollable mutants are all located at the very top of the linker, that is further away from the membrane domain compared to other tested residues (*Figure 2D*).

## Photocontrol of GluD2-A654T-I677C tethered with MAGu

We then selected the best mutant GluD2-A654T-I677C for further characterization. Because pentamidine does not fully block GluD2, even at saturating concentrations (*Williams et al., 2003*), we quantified the extent of photoswitching by blocking leak current completely, using impermeant N-Methyl-D-glucamine (NMDG). We found that MAGu blocked about 33% of the leak current in its *trans* form (*Figure 3A*). Photoregulation was fully reversible over many cycles of 380 and 535 nm light (*Figure 3B*), in agreement with the fact that azobenzenes photobleach minimally (*Beharry and Woolley, 2011*). Under our illumination conditions, light pulses of 200 ms were sufficient to fully unblock the current, while shorter illumination times could be used to finely tune the degree of blockade (*Figure 3C*). Once in the *cis* configuration, MAGu relaxes back to its thermodynamically stable *trans* state slowly, with a half-life of about 20 min in solution (*Figure 1—figure supplement 1B*). Accordingly, relief of blockade persisted for many seconds in darkness after a brief flash of 380 nm light (*Figure 3D*), eliminating the need for constant illumination, an important feature for future neurophysiology experiments.

From a pharmacological point of view, current blockade occurred in the *trans* state (535 nm) and was relieved in the *cis* configuration (380 nm) for all membrane potential tested, with very little voltage-dependence (*Figure 4A*), which contrasts with the profound voltage-dependence of block observed with pentamidine. This suggested to us that the positive charge of MAGu may not sense the electrical field of the membrane as much as pentamidine does, and thus that the two molecules may bind to different sites. To investigate whether MAGu and pentamidine compete for the same binding site, we evaluated the dose-response relationship of pentamidine block on GluD2-A654T-I677C conjugated with MAGu, under both 380 and 535 nm light (*Figure 4B*). We found the IC50s under both wavelengths to be virtually indistinguishable, favoring the idea that MAGu and pentamidine have distinct, non-overlapping binding sites. To get further molecular insight into *trans* MAGu-induced reduction of current, we performed molecular modeling experiments. After inserting the cysteine mutation, *trans* MAGu was docked by covalent docking, that is the reactive maleimide moiety was forced to be in contact with the cysteine while the rest of the molecule was free to move. We found that the guanidinium headgroup of *trans* MAGu couldn't reach the membrane-embedded lumen of GluD2 (*Figure 4C*), in agreement with our electrophysiology data. The effect of *trans* MAGu on the ion current was tested with the MOLEonline webserver (*Pravda et al., 2018*), which allowed to compute the geometry of the ion channel. We found that the photoswitch has a direct steric effect on the size of the cavity above the channel, as shown by the comparison of the computed channel in presence or absence of the photoswitch (*Figure 4D*). In addition, the charge of the photoswitch could modify the electrostatic potential in the cavity and thereby affect ion transfer.

## Optical control of the native GluD2 channel

We next sought to determine whether the non-Lc, native channel could be photocontrolled after installation of MAGu on the cysteine-substituted GluD2-I677C receptor. In heterologous expression system, activation of mGlu1 using the selective agonist 3,5-Dihydroxyphenylglycine (DHPG) was reported to trigger opening of GluD2 receptors (*Ady et al., 2014*; *Dadak et al., 2017*). Therefore, we co-expressed the b isoform of mGlu1, which displays low basal activity (*Prézeau et al., 1996*), together with GluD2 in HEK cells. Cells were labeled with MAGu and DHPG currents were recorded

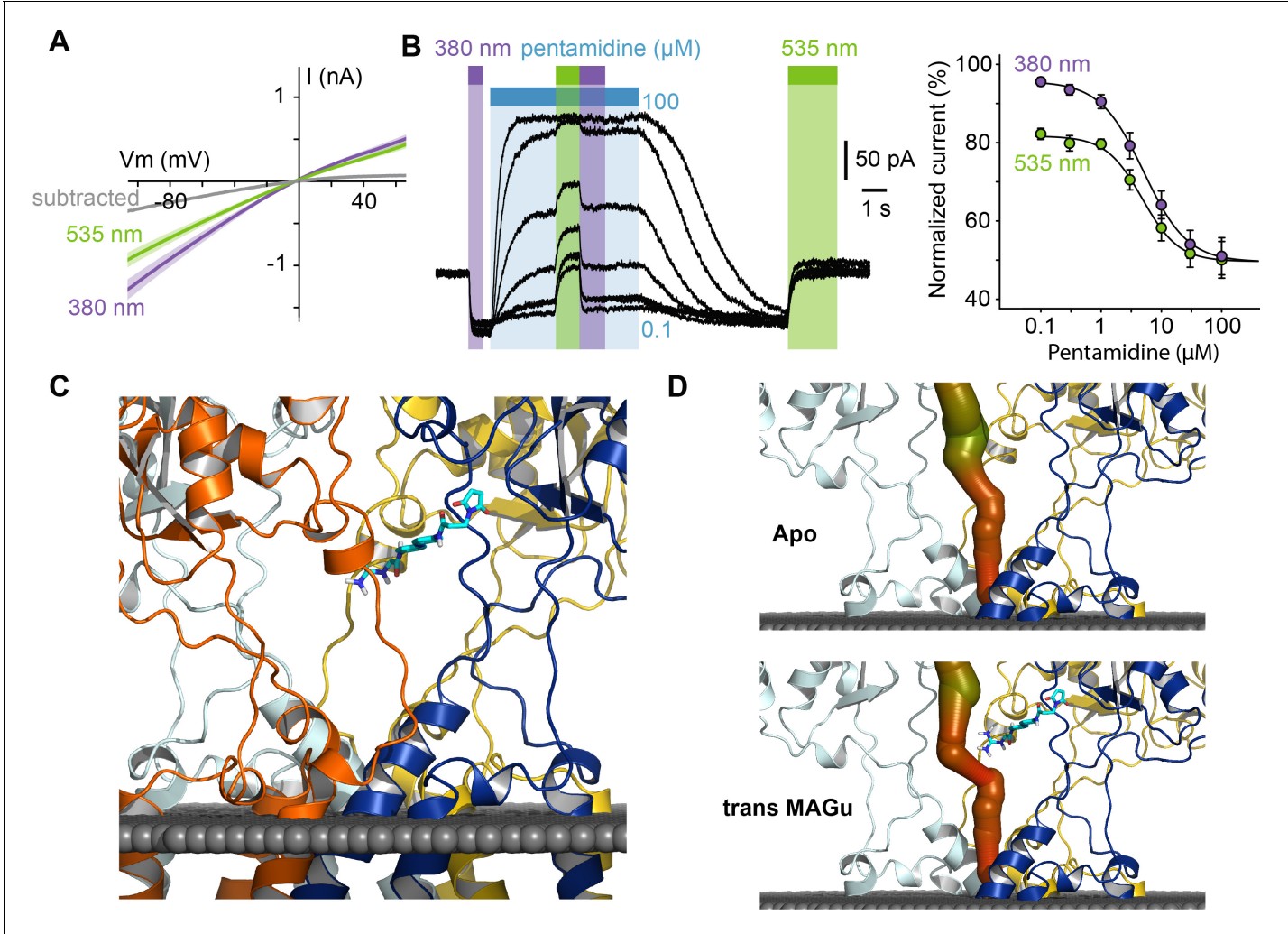

**Figure 4.** Pharmacological action of MAGu at GluD2-A654T-I677C. (**A**) Average current-versus-voltage relationship under 380 and 535 nm light (n = 21). S.E.M is shown is shade, and subtracted current in gray. (**B**) Left, representative dose-dependent blockade of the current upon pentamidine (0.1–100 μM) application, under 380 and 535 nm light. Right, quantification of pentamidine blockade under both wavelengths of light. IC50$_{380}$ = 5.1 ± 0.3 μM, IC50$_{535}$ = 4.8 ± 0.5 μM (n = 6 cells). (**C**) Molecular modeling showing *trans* MAGu tethered to I677C. (**D**) Molecular modeling showing the ion channel computed in the absence (top) and presence (bottom) of *trans* MAGu. The channel is represented with a color coding of the diameter in order to facilitate observation of the change induced by *trans* MAGu (from green, large, to red, small). One subunit is omitted for clarity. Data are presented as mean value ± sem. Source files of individual data points used for the quantitative analysis are available in the *Figure 4—source data 1*. The online version of this article includes the following source data for figure 4:

**Source data 1.** Related to *Figure 4A and B*.

while alternating between 380 and 535 nm light. We found that DHPG-induced currents were reversibly reduced by about 23% under 535 nm compared to 380 nm light for I677C, indicating that optical blockade with MAGu could be transposed to the native, non-Lc GluD2 (*Figure 5A*). Importantly, DHPG-induced currents were identical in both wavelengths of light for the WT receptor (*Figure 5B*), confirming that the effect of light is specific to the attachment of MAGu to I677C (*Figure 5C*). In addition, we observed that the holding current increased when switching from darkness to 380 nm light for I677C, and decreased when switching back to 535 nm light, but remained constant in both wavelengths of light for WT (*Figure 5D*). This suggests that a fraction GluD2 receptors are constitutively open prior to DHPG application, likely due to some basal mGlu1 activity in these cells. Altogether, these results show that the GluD2 I677C mutant labeled with MAGu (a.k.a. LiGluD2) possesses a functional ion channel, which can be gated through the mGlu signaling pathway, and which can be reversibly blocked and unblocked with green and purple light, respectively.

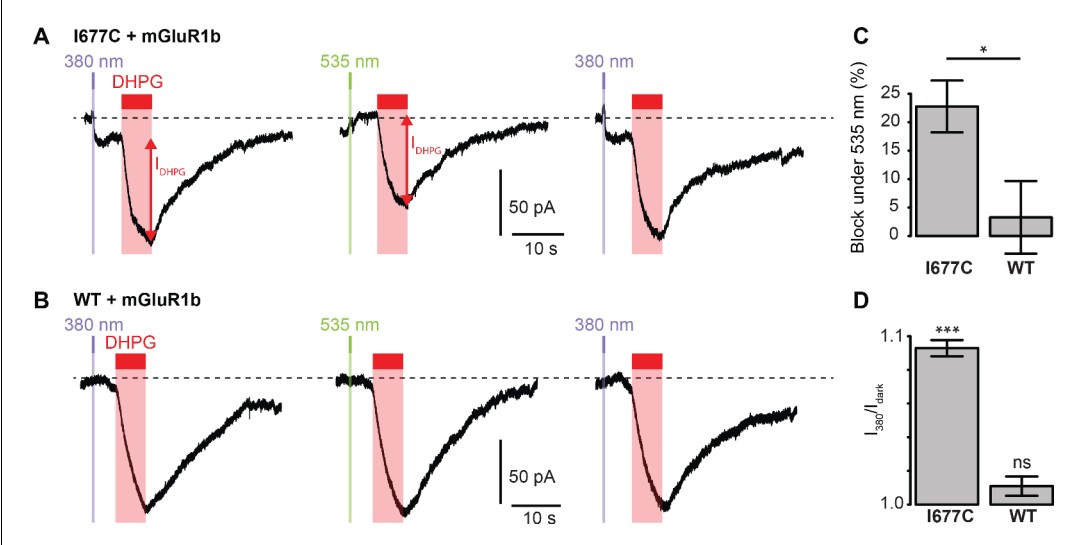

**Figure 5.** Photocontrol of GluD2-I677C (LiGluD2). (**A**) Representative DHPG-induced current (measured as indicated by the red arrow) for a MAGu-treated (20 µM, 20 min) cell co-expressing mGlu1b and GluD2-I677C, under 380, 535 and 380 nm light. Note the drop in holding current at the onset of the 380 nm illumination, and the return to the baseline under 535 nm light. (**B**) Representative DHPG-induced current for a MAGu-treated cell co-expressing mGlu1b and WT GluD2, under 380, 535 and 380 nm light. (**C**) DHPG-induced currents were reduced under 535 compared to 380 nm light for I677C (22.8 ± 4.6%, n = 8 cells) but not for WT GluD2 (3.3 ± 6.4%, n = 6 cells, p=0.02, two-sample t-test). (**D**) Ratio of the holding current recorded under 380 nm light and in darkness is different from one for I677C (1.09 ± 0.004%, p=2.44 e-7, n = 8 cells) but not for WT GluD2 (1.01 ± 0.006%, p=0.10, n = 8 cells, one-sample t-test). Source files of individual data points used for the quantitative analysis are available in the *Figure 5—source data 1*. The online version of this article includes the following source data and figure supplement(s) for figure 5:

**Source data 1.** Related to *Figure 5C and D*.

**Figure supplement 1.** MAGu does not photosensitize native GluD and GluA currents in Purkinje cells.

In order to evaluate the usefulness of LiGluD2 in neurons, we verified that MAGu treatment does not lead to photosensitization of native glutamate currents in PCs of WT mice. Since GluD2 is enriched at the parallel fiber-PC synapse (*Landsend et al., 1997*), we used local application of DHPG (200 µM) to induce inward current in both MAGu- and vehicle-treated slices. We found that the DHPG current amplitude remained unchanged under 535 nm compared to 380 nm in the two treatment conditions, resulting in a ratio of current amplitude $I_{380}/I_{535}$ measured for each cell not significantly different from 1 (*Figure 5—figure supplement 1A*). We then recorded AMPA-mediated excitatory post-synaptic currents (EPSCs) in both vehicle- and MAGu-treated PCs. We found that the amplitude of electrically evoked EPSCs was stable under 535 nm compared to 380 nm, and that the ratio of EPSC amplitudes $I_{380}/I_{535}$ was not significantly different from one in both conditions (*Figure 5—figure supplement 1B*). These control experiments demonstrate that wild-type GluD and GluA receptors, which lack a properly-positioned cysteine residue near the pore lumen, remain insensitive to light after MAGu treatment.

## Discussion

The PTL strategy has been successfully applied to several members of the iGluR family, including kainate (*Volgraf et al., 2006*) and NMDA (*Berlin et al., 2016*) receptors. In these former studies, the photoswitches were made with a glutamate head group, and were tethered to the LBD in proximity to the glutamate binding pocket, providing photocontrol of channel gating (*Reiner et al., 2015*). Because their activation mechanism is still unknown, we adopted a different strategy for photocontrolling GluD receptors. We installed the photoswitchable ligand MAGu in proximity to the pore lumen, in hope to alter ion conduction through non-competitive antagonism. We found several cysteine mutants for which current was specifically modulated by light after attachment of MAGu, notably I677C (a.k.a. LiGluD2). In LiGluD2, *trans* MAGu likely does not reach the pore lumen as originally

designed. Nevertheless, it reversibly modulates current through the open GluD2 channel with high temporal and pharmacological precision.

The compounds traditionally used to probe the ionic function of GluDs, such as pentamidine and NASPM (*Kohda et al., 2000*; *Williams et al., 2003*), are not specific of GluD and also block NMDA and AMPA receptors. As to D-serine and glycine, they partially inhibit GluD2$^{Lc}$ and mGlu1-gated GluD currents (*Naur et al., 2007*; *Ady et al., 2014*; *Benamer et al., 2018*), but they are also co-agonists of NMDA receptors. Here, the pharmacological specificity of LiGluD2 is exquisite: after MAGu treatment, only the I677C mutant, and not the WT receptor, became sensitive to light. Likewise, MAGu did not photosensitize other WT glutamate receptors expressed in native brain tissue, in agreement with previous report demonstrating that MAGu has no off-target effects on WT GABA receptors, glutamate receptors and voltage-gated ion channels (*Lin et al., 2015*). Indeed, even though the maleimide group of MAGu reacts in principle with any extracellular cysteine freely accessible on the cell surface, the ability of the tethered ligand (here guanidinium) to reach a particular site of action on any given protein in one configuration, but not the other (e.g. *trans* but not *cis*), is highly improbable. In fact, the PTL approach has already demonstrated exquisite pharmacological specificity for a large variety of cysteine-substituted ion channels and receptors (*Paoletti et al., 2019*; *Mondoloni et al., 2019*), even in complex biological settings such as brain slices (*Berlin et al., 2016*) or intact neuronal circuits in vivo (*Lin et al., 2015*; *Durand-de Cuttoli et al., 2018*).

Attachment of MAGu to GluD2 requires a single amino acid substitution, which is unlikely to disrupt the function of the receptor. In line with this, we found that the functional coupling of GluD2 with mGlu1 signaling (*Ady et al., 2014*; *Dadak et al., 2017*) was intact in LiGluD2. This enabled us to validate that activation of mGlu1 triggers the opening of the GluD2 channel in heterologous expression system, in support of earlier evidence that opening of the ion channel of GluD receptors can be triggered in response to metabotropic signaling mechanisms (*Ady et al., 2014*; *Dadak et al., 2017*; *Benamer et al., 2018*; *Gantz et al., 2020*). Even though light-induced blockade in LiGluD2 is partial, the rapid kinetics of block/unblock, coupled to the genetic specificity of the methodology, provide a unique opportunity to detect even small variations in GluD2 current, such as the tonic current we observed in heterologous expression system. LiGluD2 remains to be deployed in neuronal setting, yet we believe it will be a crucial tool for probing the ionotropic contribution of this orphan receptor to synaptic physiology.

# Materials and methods

## Key resources table

| Reagent type (species) or resource | Designation | Source or reference | Identifiers | Additional information |
|---|---|---|---|---|
| Cell line (*H. sapiens*) | HEK tsa201 | Sigma-Aldrich #96121229 | RRID:CVCL_2737 | |
| Chemical compound, drug | MAGu | 10.1016/j.neuron.2015.10.026 | | Originally named PAG-1c. Custom-synthetized by Enamine, Ukraine |
| Chemical compound, drug | DMSO | Sigma-Aldrich | D2650 | |
| Chemical compound, drug | Pentamidine | Sigma-Aldrich | 1504900 | |
| Chemical compound, drug | N-methyl-d-glucamine (NMDG) | Sigma-Aldrich | M2004 | |
| Chemical compound, drug | (R,S)−3,5-DHPG | Hello-bio | HB0026 | |
| Chemical compound, drug | CNQX | Hello-bio | HB0205 | |
| Chemical compound, drug | D-APV | Hello-bio | HB0225 | |

*Continued on next page*

*Continued*

| Reagent type (species) or resource | Designation | Source or reference | Identifiers | Additional information |
|---|---|---|---|---|
| Chemical compound, drug | SR 95531 (Gabazine) | Hello-bio | HB0901 | |
| Chemical compound, drug | CGP 55845 | Hello-bio | HB0960 | |
| Gene (*Mus musculus*) | Grid2 (glutamate receptor, ionotropic, delta 2) | Genbank | GeneID: 14804 | |
| Gene (*Rattus norvegicus*) | Grm1 (glutamate receptor, metabotropic 1) | Genbank | Gene ID: 24414 | |
| Transfected construct (*Mus musculus*) | pcDNA3-GluD2 | https://doi.org/10.1002/embr.201337371 | NM_008167.3 | |
| Transfected construct (*Mus musculus*) | pRK5-mGlu1b | Laurent Prezeau (IGF, Montpellier, France). | NM_001114330.1 | |
| Software, algorithm | R Project for Statistical Computing | http://www.r-project.org/ | RRID:SCR_001905 | |
| Software, algorithm | Modeller 9.19 | https://salilab.org/modeller/9.19/release.html | RRID:SCR_008395 | |
| Software, algorithm | Smina | https://sourceforge.net/projects/smina/ | | |

## Chemicals

Bio-grade Chemicals products was provided by Sigma-Aldrich from Merck. MAGu was synthesized as previously described (*Lin et al., 2015*) and provided by Enamine Ltd., Kyiv, Ukraine (www.enamine.net). MAGu was stored at −80°C as stock solutions in anhydrous DMSO.

## Spectrophotometry

UV-visible spectra were recorded on a Nanodrop 2000 (Thermo Scientific, 1 mm path) with 100 µM MAGu in PBS pH 7.4 (10% final DMSO). The sample was illuminated for 1 min using ultra high-power LEDs (Prizmatix) connected to an optical fiber (URT, 1 mm core, Thorlabs), followed by an immediate measurement of absorbance. Light intensity at the tip of the 1 mm fiber was 100 mW for the 390 nm LED, and 150 mW for the 520 nm LED.

## Molecular biology

The single-cysteine mutations of GuD2 were generated by site-directed mutagenesis using the Quick Change II kit (Agilent technology) performed on pcDNA3-GluD2 (*Ady et al., 2014*). All mutants were verified by sequencing.

## Cell line

We used human Embryonic Kidney cells (HEK tsA201, Sigma-Aldrich # 96121229). Cells were certified by Sigma-Aldrich. Mycoplasma contamination status were negative.

## Cell culture

Cells were cultured in 25 cm$^2$ tissue culture flask (Falcon, Vented Cap, 353109) with a culture medium composed of Dulbeco's Modified Eagle Medium (Gibco life technologies, 31966047) containing Glutamax and supplemented with Fetal Bovine Serum (10%, Gibco life technologies, 10500064), Nonessential Amino-Acids (1%, Life Technologies, 11140–035), ampicillin, streptomycin (50,000 U, Gibco, life technologies, 15140–122) and mycoplasma prophylactic (2.5 mg, InvivoGen) antibiotics.

## Transfection

HEK tsA201 cells were freshly seeded and plated out in a 6-well plate, on coverslips (10 mm) treated with poly-L-lysine hydrobromide (Sigma, P6282-5MG). Cells were transiently transfected using calcium-phosphate precipitation, as described in *Lemoine et al., 2016*, using 1 µg of cDNA of GluD2

cysteine mutant per well. For co-transfection experiments, we used mGlu1b/GluD2 ratio from 0.7 to 1, with a maximum of 2 μg of total DNA. The plasmid pRK5-mGlu1b used in this study is a generous gift of L. Prezeau (IGF, Montpellier).

## In vitro electrophysiology

Electrophysiological currents were recorded on HEK tsA201 cells at room temperature (21–25°C), 24–48 hr after transfection. Prior to whole-cell patch-clamp experiments, cells were incubated for 20 min with an extracellular solution containing 20 μM MAGu, and then washed for at least 5 min with a fresh external solution. Cells were perfused with an external solution containing (in mM): 140 NaCl, 2.8 KCl, 2 $CaCl_2$, 2 $MgCl_2$, 12 glucose, 10 HEPES and NaOH-buffered at pH 7.32. The external NMDG solution contained (in mM): 140 NMDG, 2.8 KCl, 2 $CaCl_2$, 2 $MgCl_2$, 12 glucose, 10 HEPES and was KOH-buffered at pH 7.32. Cells were patched with a borosilicate pipette (4–5 MΩ) containing an intracellular solution containing (in mM): 140 KCl, 5 $MgCl_2$, 5 EGTA, 10 HEPES, and pH-adjusted to 7.32 with KOH. For recording metabotropic activation of GluD2 by mGlu1, the internal solution contained (in mM): 140 K-gluconate, 6 KCl, 12.6 NaCl, 0.1 CaCl2, 5 Mg-ATP, 0.4 Na-GTP, 1 EGTA, 10 HEPES, and was adjusted to pH 7.32 with KOH. Pentamidine and NMDG solutions were applied using a fast-step perfusion system equipped with three square tubes (SF77B, warning instruments), as described in *Lemoine et al., 2016*. Illumination was carried out using a high-power LED system (pE-2, Cooled) mounted directly on the epifluorescence port of a vertical microscope (Slice-Scope Pro 6000, Scientifica). Light output at the focal plane was 5 and 11.7 mW/mm$^2$ for the 380 and 535 nm LEDs, respectively. Currents were recorded with an axopatch 200B and digitized with a digidata 1440 (Molecular devices). Signals were low-pass filtered (Bessel, 2 kHz) and collected at 10 kHz using the data acquisition software pClamp 10.5 (Molecular Devices). Electrophysiological recordings were extracted using Clampfit (Molecular Devices) and analyzed with R.

## Slice electrophysiology

Animal breeding and euthanasia were performed in accordance with European Commission guidelines and French legislation (2010/63/UE) and procedures were approved by the French Ministry of Research (Agreement APAFIS#16198–2018071921137716 v3). Mice at age P30-40 were anesthetized with isoflurane and decapitated. Cerebella were rapidly extracted and transferred into ice-cold ACSF supplemented with 50 mM sucrose and 1 mM kynurenic acid. The composition of ACSF in mM was as follows: 126 NaCl, 26 $NaHCO_3$, 2.5 KCl, 1.25 $NaH_2PO_4$, 1 $MgCl_2$, 2 $CaCl_2$, 20 glucose. pH was adjusted to 7.4 by continuous gassing with carbogen. Sagittal slices (250 μm) were sectioned from the vermis on a vibratome (Leica VT 1200S) and transferred to oxygenated ACSF. Slices were incubated for 15 min at 30°C then transferred at room temperature before recording. Unless stated otherwise, all the steps were performed at room temperature. Purkinje cells were visually identified using infrared Dodt contrast imaging with a 60 × water immersion objective. Whole-cell recordings from Purkinje cells in cerebellar lobules IV-VI (voltage-clamped at −70 mV, liquid junction potentials not corrected) were performed with borosilicate glass pipettes (WPI, 2–4 MΩ) pulled with a horizontal micropipette puller (Sutter instruments). Internal pipette solutions contained (in mM): 140 Cs-gluconate, 5 CsCl, 2 $MgCl_2$, 0.5 EGTA, 2 Na-ATP (pH 7.3, adjusted with CsOH). Whole-cell currents were recorded at 20 kHz and filtered with a Bessel low-pass filter at 4 kHz using a patch-clamp amplifier (Multiclamp 700B, Molecular Devices) connected to a Digidata 1440A interface board (Molecular Devices). Only Purkinje cells with a series resistance <12 MΩ (not compensated; monitored during experiments by applying 200 ms, −5 mV voltage pulses) were used for the analyses.

Slices were incubated for 20 min under continuous oxygenation with MAGu 20 μM or vehicle (DMSO 0.4%) dissolved in 750 μl ACSF in a well of 24-well plate. The slices were then washed in ACSF for at least 20 min, and transferred in the recording chamber. Photoswitching was achieved by illuminating the slice as described above alternatively at 380 nm and 535 nm for 1 s.

(RS)-DHPG (200 μM) was diluted in ACSF and locally pressure-applied using a patch pipette placed in the dendrites of the recorded Purkinje-cell. A pneumatic microinjector (Picopump, WPI) was used to deliver 0.1–0.2 ms air pressure pulses (4–10 PSI) every minute. DHPG-mediated currents were recorded at room temperature in presence of CNQX 10 μM, D-APV 25 μM, gabazine 10 μM and CGP 55845 0.5 μM. DHPG was applied immediately after the light stimulation.

Parallel fiber stimulation was achieved every 10 s with a glass pipette filled with ACSF and placed in the outer half part of molecular layer. A constant voltage isolation unit (DS3, Digitimer Ltd) was used to deliver 10 μs rectangular pulses (50–200 μA) for extracellular stimulation. Parallel fibers inputs were identified by paired-pulse facilitation at 50 ms inter stimulus interval. Amplitudes of evoked EPSCs were averaged from 6 to 12 traces. EPSCs were recorded at 30°C in presence of gabazine 10 μM. Electrical stimulations were performed immediately after the light stimulation.

### Molecular modeling

The model of the GluD receptor has been obtained by homology modeling using the software modeller version 9.19 (*Webb and Sali, 2016*). The template was that of the glutamate receptor GluA2 (PDB code 5weo) (*Twomey et al., 2017*). The automodel class has been used with slow level of MD refinement and the optimization has been repeated three times for each model. 500 models were prepared and the best, as assessed by the DOPE score, was retained for further studies.

The structure of the protein and ligand were converted to pdbqt files with the software open babel 2.4.1. Covalent docking was then performed with the software smina (*Koes et al., 2013*). The box of 25*25*25 angstrom was defined manually to encompass the mutated residue and extend to the axis of symmetry. Covalent docking forced the maleimide to be in direct contact with the SG atom of the cysteine with which it is shown experimentally to form a covalent bond. The geometry of the ion channel has been computed with MOLEonline webserver, with the 'pore' mode. The resulting ion channel was color-coded as a function of the diameter of the channel allowing to illustrate the reduction of the diameter from a large (green) to a small (red) diameter.

### Data analysis

Data are plotted as mean ± SEM. Total number (n) of cells in each group and statistics used are indicated in figure and/or figure legend. Comparisons between means were performed using parametric tests (two-sample t-test, Normality always verified, Shapiro-Wilk test of normality). Homogeneity of variances was tested preliminarily and the t-tests were Welch-corrected accordingly. For comparison with theoretical values of 0 or 1, we performed either one-sample t-tests when Normality was verified, or a non-parametric test (one-sample Wilcoxon tests) when Normality was not verified. [#]$p<0.1$, *$p<0.05$, **$p<0.01$, ***$p<0.001$.

Time-course of current unblock and of thermal relaxation were fitted with the following mono-exponential function:

$$y = 100 \times \left(1 - e^{-kx}\right) \qquad (1)$$

with k the decay constant, and ln2/k the half-life.
Dose-response relationships were fitted with the following equation:

$$y = MIN + \frac{MAX - MIN}{1 + \left(\frac{x}{IC50}\right)^{nH}} \qquad (2)$$

with MAX the maximal current, MIN the minimal current, IC50 the pentamidine concentration yielding half block, and $n_H$ the Hill number.

## Acknowledgements

Authors would like to thank Nadine Mouttajagane and Manel Badsi for their help with molecular biology work. This work was supported by funding provided by the French Agency for Research (ANR-16-CE16-0014-01 to LT, ANR-11-LABX-0011 to AT), by the 'Initiative d'Excellence' (cluster of excellence LABEX Dynamo) to AT, by the Foundation for Medical Research (FRM, Equipe FRM EQU201903007961 to PF) and by a post-doctoral fellowship from the Labex BioPsy to DL.

## Additional information

### Funding

| Funder | Grant reference number | Author |
|---|---|---|
| Agence Nationale de la Recherche | ANR-16-CE16-0014-01 | Ludovic Tricoire |
| Fondation pour la Recherche Médicale | FRM EQU201903007961 | Philippe Faure |
| LABEX Dynamo | | Antoine Taly |
| Agence Nationale de la Recherche | ANR-11-LABX-0011 | Antoine Taly |
| LABEX Biopsy | | Damien Lemoine |

The funders had no role in study design, data collection and interpretation, or the decision to submit the work for publication.

### Author contributions

Damien Lemoine, Data curation, Formal analysis, Writing - review and editing; Sarah Mondoloni, Data curation, Visualization, Writing - review and editing; Jérome Tange, Data curation; Bertrand Lambolez, Supervision, Writing - review and editing; Philippe Faure, Supervision, Funding acquisition, Writing - review and editing; Antoine Taly, Data curation, Funding acquisition, Visualization, Writing - review and editing; Ludovic Tricoire, Data curation, Supervision, Funding acquisition, Writing - review and editing; Alexandre Mourot, Conceptualization, Formal analysis, Supervision, Visualization, Writing - original draft

### Author ORCIDs

Sarah Mondoloni https://orcid.org/0000-0002-6134-3715
Bertrand Lambolez http://orcid.org/0000-0002-0653-480X
Philippe Faure http://orcid.org/0000-0003-3573-4971
Antoine Taly https://orcid.org/0000-0001-5109-0091
Ludovic Tricoire https://orcid.org/0000-0003-3345-1468
Alexandre Mourot https://orcid.org/0000-0002-8839-7481

### Decision letter and Author response

Decision letter https://doi.org/10.7554/eLife.59026.sa1
Author response https://doi.org/10.7554/eLife.59026.sa2

## Additional files

### Supplementary files

• Transparent reporting form

### Data availability

All data generated or analysed during this study are included in the manuscript and supporting files. Source data have been provided for all the figures.

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
