## [Decision Letter]

**Acceptance summary:**

GluD2 is a member of the ionotropic glutamate receptor family, but its specific role in the central nervous system is not yet understood. This study demonstrates a novel chemogenetic tool to probe function of GluD2 in HEK cells. This tool has the potential to be very powerful to advance the understanding of GluD2 channel function in neurons since it appears highly selective.

**Decision letter after peer review:**

Thank you for submitting your article "Probing the ionotropic activity of the orphan glutamate delta 2 receptor with genetically-engineered photopharmacology" for consideration by *eLife*. Your article has been reviewed by three peer reviewers, and the evaluation has been overseen by a Reviewing Editor (Merritt Maduke) and Kenton Swartz as the Senior Editor. The following individuals involved in review of your submission have agreed to reveal their identity: Stephanie C Gantz (Reviewer #1); Lonnie Wollmuth (Reviewer #2); Anna Koster (Reviewer #3).

The reviewers have discussed the reviews with one another, and the Reviewing Editor has drafted this decision to help you prepare a revised submission.

The editors have judged that your manuscript is of interest; however, as described below, additional experiments are required to support the conclusions. Therefore, we would like to draw your attention to changes in our revision policy that we have made in response to COVID-19 (https://elifesciences.org/articles/57162). In recognition that many researchers have temporarily lost access to the labs, we will give authors as much time as they need to submit revised manuscripts.

Summary:

Unlike other ionotropic glutamate receptors, GluD2 is not gated by glutamate. No specific or high-affinity chemical modulators that induce channel activity exist for this receptor. To address this challenge, the authors used a previously characterized photoswitchable tethered ligand (PTL) called MAGu to target a very non-specific blocker (pentamidine) to a new ion channel target (the GluD2 receptor). This approach (using this exact PTL) has been used to target knock-in cysteine mutants of the GABAA receptor in mouse brain slices and in vivo in an awake, behaving mouse. Based on this precedent, it is not unreasonable to believe that this tool could similarly be used for the GluD2 receptor, which would be a significant advance in the field for understanding the physiological role of this protein in disease. However, there are concerns about signal-to-noise, since the pore block by trans-MAGu is only a fraction of total presumed current through GluD; therefore, it has not yet been demonstrated that the MAGu response will be sufficient for physiological studies.

Essential revisions:

1) In order to evaluate the potential of this new tool, it is essential that the authors provide a more thorough quantification of its effects, as described below, which will be relevant to the signal:noise in physiological experiments.

The photo-switchable capability of MAGu is convincing, but what is not clear is how good of a blocker trans-MAGu is. It is clear that trans-MAGu is not producing full block of leak current, but it is not quantified. These data need to be added to have a clear understanding of the strategy and mechanism. cis-MAGu unblocks a portion of leak current, but it is not clear how much of the total leak current is blocked by trans-MAGu. Knowing how much leak current is blocked by trans-MAGu is critical to interpreting the effect on mGluR-activated GluD2 current.

In Figure 2C I677C, there is still substantial leak in 535 nm. The quantification in Figure 2C (% photoswitching) shows the % of I-Blockphoto over I-Blockpenta, but the arrows in the righthand trace, it would appear I-Blockphoto is actually the current unblocked. It would be helpful to quantify the amount of leak current blocked by trans-MAGu. In Figure 2C, the extent of block for photoswitching is being quantified relative to that for pentamidine, which is reasonable. However, for pentamidine, what is the concentration used for the experiments? Where is it at on the concentration-block curve for pentamidine? Presumably, if complete block the leak current should go to zero and hence the efficacy of block of photoswitching blocker would be less (e.g., Figure 4B). Please clarify. Additional discussion as the structural basis for incomplete block may also be helpful.

How does MAGu work on the cysteine-engineered receptor that would presumably be used for future in vivo studies? Because the GluD2-I677C point mutant (lacking the L654T background) does not show current, the authors use the known effect of mGlu1 receptor agonism as a readout of GluD2-I677C activity in response to light and only see a 23% decrease in mGlu1 current--is this very small effect physiologically significant or to be expected? It seems like MAGu might be a useful tool to modulate GluD2 in Lurcher mice (which harbor the L654T mutation), but it is hard to know what the probe efficacy and usefulness is for evaluating the physiology of the WT GluD2 receptor in the absence of a way to measure a direct functional effect on the channel. How else might this be addressed?

Discussion paragraph two states that the WT receptor is insensitive to MAGu, but it is not clear where those data are presented. It would be beneficial to show the magnitude of the DHPG-induced current in WT GluD2-expressing cells before and after addition of MAGu to address the possibility that MAGu affects the current irrespective of trans- or cis- conformation.

It is also not clear how MAGu will be selective for site-specific conjugation when introduced in a neuronal setting. Is it expected MAGu will react with any available cysteine? It would be helpful to discuss possible limitations going forward towards use in neurons.

2) The Introduction and Abstract are rather general and antiquated, to the disservice of the readers. It may be time to move away from the notion that ion channel function of GluD is debated. The authors have published many elegant studies demonstrating ion channel function. By appearances of the literature, the interpretation of these studies are not contested. In addition to pharmacology, ion channel function of GluD has been demonstrated using selective genetic strategies (e.g. Ady et al., 2013; Benamer et al., 2018; Gantz et al., 2020). To this end, the Abstract and Introduction should be changed. It does not seem fitting to state "direct evidence for ionotropic activity of GluD in neuronal setting [sic] is lacking" provided the studies referenced above. Broadly, the readers would benefit from restructuring of the Introduction and Abstract to state the specific issue addressed by the present study (i.e. the lack of specific antagonists/pore blockers to study GluD without affecting other iGluRs) and highlight the potential application of the ligand.

3) It would be helpful to define early and explicitly what the photoswitchable functional strategy is – that it is working via a pore block mechanism. In the Abstract, for example, instead of calling it “…a photoswitchable ligand.” how about just “…a photoswitchable pore blocker." The functional strategy – that you are generating a photoswitchable pore blocker – should also be explicitly stated in the Introduction, where right now it is touched on but not explicitly stated.

4) PTLs have been shown to generate a high local concentration of ligand to accelerate pharmacological response (and in this case, provide some level of specificity for a very non-specific, greasy cation), but it is hard to rationalize "absolute" pharmacological specificity claimed by the authors (Abstract, Discussion paragraph two). At the mid-μM concentrations required to elicit response, it seems unlikely that MAGu will not react with any other extracellular cysteines present in cells. Further, the guanidinium group by itself will certainly not direct the maleimide reactivity towards GluD2 over any other cation channel or electronegative protein surface. The language of this claim should be modified in the absence of other types of specificity assays.

5) Figure 4A. Please also show the difference current and contrast/compare to what is shown in Figure 2A. This would clarify the “voltage-independence” of action for those unfamiliar.

6) Figure 4D. It is not clear how the “ion channel” or red/green pore was generated. Is this from the structure or from some modeling? Please add details. This is an interesting figure, but it appears somewhat speculative and requires more details for the reader to understand its basis. What is driving the positioning of the trans MAGu? Is it being fixed? And what is driving the change in the coloration – presumed pore blocking by trans MAGu?

[Editors' note: further revisions were suggested prior to acceptance, as described below.]

Thank you for submitting your article "Probing the ionotropic activity of glutamate GluD2 receptor in HEK cells with genetically-engineered photopharmacology" for consideration by *eLife*. Your article has been reviewed by the three original peer reviewers, and the evaluation has been overseen by Merritt Maduke as the Reviewing Editor and Kenton Swartz as the Senior Editor. The following individuals involved in review of your submission have agreed to reveal their identity: Stephanie C Gantz (Reviewer #1); Lonnie Wollmuth (Reviewer #2); Anna Koster (Reviewer #3).

The reviewers have discussed the reviews with one another. They agree the revisions have greatly strengthened the manuscript but that a few additional relatively minor revisions are essential. The Reviewing Editor has drafted this decision to help you prepare a revised submission.

Summary:

Unlike other ionotropic glutamate receptors, GluD2 is not gated by glutamate. No specific or high-affinity chemical modulators that induce channel activity exist for this receptor. To address this challenge, the authors used a previously characterized photoswitchable tethered ligand (PTL) called MAGu to target a very non-specific blocker (pentamidine) to a new ion channel target (the GluD2 receptor). This approach (using this exact PTL) has been used to target knock-in cysteine mutants of the GABAA receptor in mouse brain slices and in vivo in an awake, behaving mouse. Based on this precedent, it is not unreasonable to believe that this tool could similarly be used for the GluD2 receptor, which would be a significant advance in the field for understanding the physiological role of this protein in disease. However, there are concerns about signal-to-noise, since the pore block by trans-MAGu is only a fraction of total presumed current through GluD; therefore, it has not yet been demonstrated that the MAGu response will be sufficient for physiological studies.

Revisions:

It is unclear how the block of DHPG-induced current was measured, especially in regards to the tonic current. Figure 5A top-left shows the tonic current and then the additional DHPG-induced current. The dashed line makes it seem that this is where the magnitude of the DHPG-induced current was measured from. But it would be best to measure DHPG-induced current as a change from the tonic inward current, which is perhaps what the red arrow is indicating? Including details in the figure legend would be helpful.

If the inhibition of DHPG-induced current included the tonic current, it may be substantially less than 22% block. If this is the case, we would suggest paired statistics to be sure the decrease in DHPG-current is significant, especially since there is some variability in both the mutant and WT conditions (seen in source data).

---

## [Author Response]

Essential revisions:1) In order to evaluate the potential of this new tool, it is essential that the authors provide a more thorough quantification of its effects, as described below, which will be relevant to the signal:noise in physiological experiments.The photo-switchable capability of MAGu is convincing, but what is not clear is how good of a blocker trans-MAGu is. It is clear that trans-MAGu is not producing full block of leak current, but it is not quantified. These data need to be added to have a clear understanding of the strategy and mechanism. cis-MAGu unblocks a portion of leak current, but it is not clear how much of the total leak current is blocked by trans-MAGu. Knowing how how much leak current is blocked by trans-MAGu is critical to interpreting the effect on mGluR-activated GluD2 current.

We understand the reviewers concern. To address this issue, we now have compared, for the best mutant (A654T-I677C), the degree of block with pentamidine (100 uM) and with transMAGu to that induced by NMDG, a large impermeant organic cation that blocks virtually all leak current (see new Figure 3A). We found that pentamidine blocks about 75% of the leak current, while trans MAGu blocks about 33% of the leak current.

In Figure 2C I677C, there is still substantial leak in 535 nm. The quantification in Figure 2C (% photoswitching) shows the % of I-Blockphoto over I-Blockpenta, but the arrows in the righthand trace, it would appear I-Blockphoto is actually the current unblocked. It would be helpful to quantify the amount of leak current blocked by trans-MAGu. In Figure 2C, the extent of block for photoswitching is being quantified relative to that for pentamidine, which is reasonable. However, for pentamidine, what is the concentration used for the experiments? Where is it at on the concentration-block curve for pentamidine? Presumably, if complete block the leak current should go to zero and hence the efficacy of block of photoswitching blocker would be less (e.g., Figure 4B). Please clarify. Additional discussion as the structural basis for incomplete block may also be helpful.

Block_photo_ was renamed Block_MAGu_ for clarity, because block occurs also in darkness, and not just under green light. In Figure 2C, current is first recorded in darkness, when MAGu is in its trans, blocking state. Illuminating with 380 nm light converts MAGu to cis and relieves blockade. The arrow in the figure has now been moved to the dark period for clarity on how we quantify Block_MAGu_.

For pentamidine, we used a saturating concentration (100 uM, see Figure 4B). This is now clearly stated both in the text and in the figure legend. Yet, even at saturating concentration, pentamidine does not block GluD fully (Figure 4B), as previously reported (Williams et al., 2003). Therefore, it makes sense indeed to quantify the degree of photoswitching by comparing the block under 535 nm light to that induced by NMDG. This is now done in Figure 3C, for the “best” mutant I677C (see response to previous comment).

In agreement with the study by Williams et al., 2003, we observed partial block of GluD2Lc current, even for saturating concentrations of pentamidine. It was suggested that pentamidine may easily permeate GluD channels. Our current manuscript does not add any new information regarding the structural basis for pentamidine block, and thus we believe adding such discussion would be too speculative.

How does MAGu work on the cysteine-engineered receptor that would presumably be used for future in vivo studies? Because the GluD2-I677C point mutant (lacking the L654T background) does not show current, the authors use the known effect of mGlu1 receptor agonism as a readout of GluD2-I677C activity in response to light and only see a 23% decrease in mGlu1 current--is this very small effect physiologically significant or to be expected? It seems like MAGu might be a useful tool to modulate GluD2 in Lurcher mice (which harbor the L654T mutation), but it is hard to know what the probe efficacy and usefulness is for evaluating the physiology of the WT GluD2 receptor in the absence of a way to measure a direct functional effect on the channel. How else might this be addressed?

The 23% decrease in mGlu1-induced current is not very different from the 33% photoswitching we are now reporting for MAGu blockade when compared to NMDG block (Figure 3A). We understand that such amount of photoswitching may appear small. However, the PTL technology has two important features that should relief the reviewers’ concerns.

First, the PTL technology is highly specific to the engineered cysteine mutation, as already demonstrated for a large variety of cysteine-substituted ion channels and receptors, including voltage-gated potassium channels, GABA receptors, metabotropic and ionotropic glutamate receptors and nicotinic acetylcholine receptors (see our recent review Paoletti et al., 2019 for a detailed list). This pharmacological specificity lies in the strong geometrical constrains between the tethered ligand and the engineered receptor, notably the strict distance between the attachment site (the cysteine mutation) and the ligand binding pocket. As a result, during the engineering step, we usually find that only one (or a handful at most) of the screened cysteine mutants can be made photocontrollable using this strategy. The PTL technology has now been deployed in complex neuronal setting, including mice and fish in vivo, with absolutely no adverse effects (see our review). Most importantly, the photoswitch MAGu we have used here has already been deployed in mice in vivo to control cysteine-substituted GABAA receptors (Lin et al., 2015), and showed absolutely no off-target effect on wild-type GABA receptors, glutamate receptors, or voltage-gated ion channels (see Figure S3 in Lin et al., 2015). We now have conducted similar controls (new Figure 5—figure supplement 1) and demonstrate in this article the specificity of our approach. We notably show that after MAGu treatment, GluD and GluA currents remain insensitive to the photostimulations, most likely because they lack a properly-positioned cysteine residue near the pore lumen.

Second, with PTLs, antagonism (or agonism) can be switched on and off within (milli)seconds, owing to the rapid kinetics of azobenzene photoswitching (see Figure 3B). This contrasts with traditional pharmacological compounds: the onset of their effect upon exposure takes a long time to develop and they can be difficult to remove, especially when they have high affinity, highly selectivity for their receptor. Hence, with PTLs, even a slight change in receptor activation efficacy can be reported rapidly with high fidelity and accuracy. This is for instance illustrated in Figure 5A: switching between 380 and 525 nm light rapidly and reversibly changes the holding current in the absence of DHPG. The light-induced current is small in amplitude, but because it can be reversibly switched on and off with high reproducibility, it unveils the existence of a tonically-activated GluD2 receptor in these cells.

For these reasons, we are confident that LiGLuD2 will be useful, in combination with a sensitive technique such as electrophysiology, to measure the direct functional effect of GluD2 photocontrol on neuronal physiology. Additionally, antagonism does not have to be complete to observe a strong physiological and/or phenotypical impact (see for instance Durand-de Cuttoli et al., 2018). The I677C mutant could be virally transduced in neuronal cells, either of the WT or the GluD2 knock-out mouse, with the idea of understanding when and where GluD2 is active. We don’t think Lurcher mice would help address this question.

These aspects of the PTL technology are now discussed in greater details in the Discussion section of the manuscript.

Discussion paragraph two states that the WT receptor is insensitive to MAGu, but it is not clear where those data are presented. It would be beneficial to show the magnitude of the DHPG-induced current in WT GluD2-expressing cells before and after addition of MAGu to address the possibility that MAGu affects the current irrespective of trans- or cis- conformation.

Our intention was to point that after MAGu treatment, the WT receptor remains insensitive to light, indicating that photocontrol is specific to the cysteine mutant. We agree with the reviewers that our sentence was misleading. We did not show that MAGu treatment per se has no effect on the WT receptor. Sentence was changed accordingly.

It is also not clear how MAGu will be selective for site-specific conjugation when introduced in a neuronal setting. Is it expected MAGu will react with any available cysteine? It would be helpful to discuss possible limitations going forward towards use in neurons.

The maleimide group of MAGu will indeed react with freely available cysteines, yet just on cell surface proteins (the highly reductive environment of the cytoplasm precludes maleimide conjugation to intracellular proteins). Nevertheless, as discussed above, this approach to photosensitizing receptors has proved to be remarkably specific in both fish and mice, with wild-type neurons or animals being unaffected by light after PTL treatment. Indeed, cysteine is a relatively infrequent amino acid (about 3%) often engaged in disulfide bonds (-S-S-) within or between polypeptide chains. Only free cysteines are available for PTL conjugation. In addition, there is a strong geometrical constrain for photocontrol: the cysteine residue has to be located at an ideal distance to the ligand binding pocket to result in photocontrol of protein activity. These limitations are now included in the Discussion.

2) The Introduction and Abstract are rather general and antiquated, to the disservice of the readers. It may be time to move away from the notion that ion channel function of GluD is debated. The authors have published many elegant studies demonstrating ion channel function. By appearances of the literature, the interpretation of these studies are not contested. In addition to pharmacology, ion channel function of GluD has been demonstrated using selective genetic strategies (e.g. Ady et al., 2013; Benamer et al., 2018; Gantz et al., 2020). To this end, the Abstract and Introduction should be changed. It does not seem fitting to state "direct evidence for ionotropic activity of GluD in neuronal setting [sic] is lacking" provided the studies referenced above. Broadly, the readers would benefit from restructuring of the Introduction and Abstract to state the specific issue addressed by the present study (i.e. the lack of specific antagonists/pore blockers to study GluD without affecting other iGluRs) and highlight the potential application of the ligand.

Even though some recent articles in the field still mention that GluD receptor do not generate ionic current, we agree that there is increasing evidence, from our group as well as from the lab of Stephanie Gantz, that GluD receptors can mediate slow excitatory current in neurons. Therefore, we modified the Introduction, and to lesser extent the Abstract, to better highlight the lack of (specific) pharmacology for GluDs, and the potential of our novel tool for the assessment of the ionotropic functions of GluDs in neurons.

3) It would be helpful to define early and explicitly what the photoswitchable functional strategy is – that it is working via a pore block mechanism. In the Abstract, for example, instead of calling it “…a photoswitchable ligand.” how about just “…a photoswitchable pore blocker." The functional strategy – that you are generating a photoswitchable pore blocker – should also be explicitly stated in the Introduction, where right now it is touched on but not explicitly stated.

We agree. We modified the text accordingly.

4) PTLs have been shown to generate a high local concentration of ligand to accelerate pharmacological response (and in this case, provide some level of specificity for a very non-specific, greasy cation), but it is hard to rationalize "absolute" pharmacological specificity claimed by the authors (Abstract, Discussion paragraph two). At the mid-μM concentrations required to elicit response, it seems unlikely that MAGu will not react with any other extracellular cysteines present in cells. Further, the guanidinium group by itself will certainly not direct the maleimide reactivity towards GluD2 over any other cation channel or electronegative protein surface. The language of this claim should be modified in the absence of other types of specificity assays.

MAGu will indeed react with other freely accessible cysteines on the cell surface, as discussed above. And because of the predictable low affinity of the guanidinium group for GluD2, we don’t expect any increase selectivity for I677C labeling. Yet, as discussed above, and as demonstrated with our supplementary figure, even though tethering is not specific to GluD2-I677C, photosensitization is. That said, we removed the word “absolute” which was somewhat excessive.

5) Figure 4A. Please also show the difference current and contrast/compare to what is shown in Figure 2A. This would clarify the “voltage-independence” of action for those unfamiliar.

We have modified Figure 4A for better comparison with Figure 2A. Instead of showing representative traces, we now show average traces + SEM for both wavelengths of light, together with the subtracted trace.

6) Figure 4D. It is not clear how the “ion channel” or red/green pore was generated. Is this from the structure or from some modeling? Please add details. This is an interesting figure, but it appears somewhat speculative and requires more details for the reader to understand its basis. What is driving the positioning of the trans MAGu? Is it being fixed? And what is driving the change in the coloration – presumed pore blocking by trans MAGu?

We added details to the Materials and methods and Results sections, to explain the modeling better. The geometry of the ion channel has been computed with MOLEonline webserver, with the “pore” mode. The resulting ion channel was color-coded as a function of the diameter of the channel allowing to illustrate the reduction of the diameter from a large (green) to a small (red) diameter. The maleimide moiety, i.e. the reactive part of MAGu, was forced to be in contact with the cysteine while the rest of the molecule was free to move.

[Editors' note: further revisions were suggested prior to acceptance, as described below.]

Revisions:It is unclear how the block of DHPG-induced current was measured, especially in regards to the tonic current. Figure 5A top-left shows the tonic current and then the additional DHPG-induced current. The dashed line makes it seem that this is where the magnitude of the DHPG-induced current was measured from. But it would be best to measure DHPG-induced current as a change from the tonic inward current, which is perhaps what the red arrow is indicating? Including details in the figure legend would be helpful.If the inhibition of DHPG-induced current included the tonic current, it may be substantially less than 22% block. If this is the case, we would suggest paired statistics to be sure the decrease in DHPG-current is significant, especially since there is some variability in both the mutant and WT conditions (seen in source data).

The magnitude of the DHPG current was not measured from the dashed line, but from the change in tonic inward current, as indicated by the red arrow in the figure indeed. Hence the block is 22%, not substantially less. This is now clearly stated in the figure legend.